# 7-*O*-Methylluteolin Suppresses the 2,4-Dinitrochlorobenzene-Induced Nrf2/HO-1 Pathway and Atopic Dermatitis-like Lesions

**DOI:** 10.3390/antiox11071344

**Published:** 2022-07-08

**Authors:** Tae-Young Kim, No-June Park, Beom-Geun Jo, Jin-Hyub Paik, Sangho Choi, Su-Nam Kim, Min Hye Yang

**Affiliations:** 1Department of Pharmacy, College of Pharmacy, Pusan National University, Busan 46241, Korea; taeyour@pusan.ac.kr (T.-Y.K.); bg_jo@pusan.ac.kr (B.-G.J.); 2Natural Products Research Institute, Korea Institute of Science and Technology, Gangneung 25451, Korea; H20508@kist.re.kr; 3International Biological Material Research Center, Korea Research Institute of Bioscience and Biotechnology, Daejeon 34141, Korea; jpaik@kribb.re.kr (J.-H.P.); decoy0@kribb.re.kr (S.C.)

**Keywords:** *Wikstroemia ganpi*, 7-*O*-methylluteolin, atopic dermatitis, antioxidant, anti-inflammation

## Abstract

7-*O*-methylluteolin (7-ML) is a flavonoid isolated from the aerial parts of *Wikstroemia ganpi* (*W. ganpi*). We describe the anti–atopic dermatitis (AD) effects of 7-ML in tert-butyl hydroperoxide (tBHP)-induced HepG2 cells and 2,4-dinitrochlorobenzene (DNCB)-induced SKH-1 hairless mice. Results demonstrated that 7-ML dose-dependently inhibited the activation of Nrf2 (nuclear factor-erythroid 2-related factor 2) in tBHP-induced HepG2 cells. 7-ML applied topically to our DNCB-induced mouse model upregulated the antioxidant protein expression (phosphorylated Nrf2 (pNrf2), Nrf2, and heme oxygenase-1 (HO-1)) in skin tissues, improved epidermal thickness, and reduced mast cell infiltration into the skin. In addition, 7-ML reduced the serum levels of immunoglobulin E (IgE) and interleukin-4 (IL-4) and improved skin barrier functions. These results suggest that 7-ML should be considered a novel antioxidant and anti-AD agent.

## 1. Introduction

Atopic dermatitis (AD) is the most common chronic skin disease and affects up to 20% of children and 3% of adults [1]. Most AD patients scratch the affected skin and cause lichenification and abrasions [2]. More fundamentally, the skin barrier is impaired, which results in skin dehydration and increased transepidermal water loss (TEWL) [3]. These symptoms are closely related to the activations of cytokines, such as interleukin-4 (IL-4) and IL-13 and immunoglobulin E (IgE) produced by T helper 2 (Th2) cells that are overexpressed in the peripheral blood of acute AD patients [3,4]. Inflammatory responses induced by these mediators produce reactive oxygen species (ROS) [5], which in turn deplete the levels of antioxidant enzymes required to maintain redox homeostasis and contribute to the pathogenesis of chronic AD by causing stratum corneum damage and activating inflammatory signaling pathways [6,7]. Therefore, as Th2-related reactions and oxidative stress play important roles in the etiology of AD, topical corticosteroids or topical calcineurin inhibitors with immunomodulatory effects are currently used to treat AD [8,9]. However, these agents are unsuitable for long-term treatment as they can cause epidermal barrier dysfunction and have side effects that include burning, itching, and skin atrophy [9].

Flavones are a class of flavonoids with three functional groups and a tricyclic backbone (C6–C3–C6) composed of A, C, and B rings [10]. These compounds have long been considered potent natural anti-inflammatory and antioxidant substances, and their in vivo effects have been suggested to be structure related [10]. Luteolin and its derived glycosides are representative examples of the flavones used to treat inflammatory diseases [11]. Luteolin has a hydroxyl group on its A and B rings and potently inhibits the effects of inflammatory mediators and oxidative stress [11,12]. Luteolin and luteolin glycosides impede the activations of inflammatory signaling pathways and enhance the expressions of antioxidant enzymes [13,14]. Moreover, methylation of the hydroxyl group of luteolin has been reported to result in potent radical scavenging activity, inhibit IgE-mediated reactions, and inhibit anti-inflammatory effects at very low concentrations [15,16,17]. Recently, several authors who investigated the antiallergic activities of flavones, including luteolin and its derivatives, suggested that luteolin might provide an effective treatment for AD [18,19,20].

*Wikstroemia ganpi (W. ganpi)* belongs to the genus *Wikstroemia* (Thymelaeaceae), which is widely distributed in Australia, Japan, and southern coastal regions of Korea [21]. The genus *Wikstroemia* is composed of around 70 species, many of which are medicinal plants that have been widely used to treat coughs, edema, pneumonia, rheumatism, and inflammatory diseases [22,23]. Several researchers have reported that extracts of members of this genus contain flavonoids, coumarins, and lignans with anti-inflammatory, antioxidant, antiviral, and antitumor properties [23,24,25]. In a previous study, we found that *W. ganpi* extract and one of its active components, 7-*O*-methylluteolin (7-ML), suppressed the expressions of some AD-related genes [26,27]. 7-ML has been isolated from diverse plants, such as *W. ganpi*, *Daphne oleoides*, *Avicennia marina*, *Coleus parvifolius*, and *Thymus vulgaris* [27,28,29,30,31], and has been reported to have anticancer [29], anti-HIV [30], and antioxidant [31] effects. The present study was undertaken to investigate the anti-AD effect of 7-ML isolated from *W. ganpi* using HepG2 cells and a DNCB (2,4-dinitrochlorobenzene)-induced mouse model of AD.

## 2. Materials and Methods

### 2.1. Plant Material

The aerial parts of *W. ganpi* (Siebold and Zucc.) Maxim. were collected in Geumsa-ri, Yeongnam-myeon, Goheung-gun, Jeollanam-do, Republic of Korea, and identified by Dr. Jin-Hyub Paik, International Biological Material Research Center, Korea Research Institute of Bioscience and Biotechnology. Plant voucher specimens (#PNU-0027) were deposited at the Medicinal Herb Garden, Pusan National University.

### 2.2. Extraction and Isolation of 7-ML from W. ganpi

Dried aerial parts of *W. ganpi* (4.27 kg) were extracted two times with 95% MeOH at room temperature for 90 min, and then the extract was evaporated in a vacuum (35–40 °C) to obtain *W. ganpi* MeOH extract (483 g). The extract was then dissolved in H_2_O and partitioned sequentially using *n*-hexane (4 L), EtOAc (4 L), and *n*-BuOH saturated with H_2_O (4 L). The EtOAc extract (41.3 g) was subjected to silica gel column chromatography and eluted using a CH_2_Cl_2_/MeOH (30:1) mixture to obtain 17 fractions (WGE 1–WGE 17). Fraction WGE 7 was recrystallized in a refrigerator and yielded crystals of 7-*O*-methylluteolin (7-ML, 310.2 mg) [32]. The structure of 7-ML was determined by NMR (nuclear magnetic resonance spectroscopy) and HRESIMS (high-resolution electrospray ionization mass spectrometry). NMR (^1^H, ^13^C, HMQC, HMBC, and NOESY) spectra were recorded on a Bruker AVANCE NEO 500 MHz spectrometer (Bruker, Billerica, MA, USA), and HRESIMS analysis was performed on an Agilent 1290 Infinity UHPLC (ultra-high performance liquid chromatography) system coupled to an Agilent 6530 Accurate-Mass Quadrupole Time-of-Flight (Q-TOF) mass spectrometer (Agilent Technologies, Santa Clara, CA, USA).

**7-*O*-methylluteolin:** Pale yellow powder; HREISMS (*m/z*) 299 [M-H]^−^; ^1^H NMR (500 MHz, DMSO-*d*_6_): δ 7.45 (H-2′, d, *J* = 2.2 Hz, 1H), 7.43 (H-6′, s, 1H), 6.90 (H-5′, d, *J* = 8.2 Hz, 1H), 6.71 (H-3, s, 1H), 6.69 (H-8, d, *J* = 2.3 Hz, 1H), 6.35 (H-6, d, *J* = 2.3 Hz, 1H), 3.86 (7-OCH_3_, s, 3H); ^13^C NMR (126 MHz, DMSO-*d*_6_): δ 181.8 (C-4), 165.1 (C-7), 164.3 (C-2), 161.2 (C-9), 157.2 (C-5), 149.9 (C-4′), 145.8 (C-3′), 121.5 (C-1′), 119.1 (C-6′), 116.0 (C-5′), 113.5 (C-2′), 104.7 (C-10), 103.1 (C-3), 97.9 (C-6), 92.6 (C-8), 56.0 (7-OCH_3_) (Figure 1).

### 2.3. Cell Culture

HepG2 cells (a human hepatoma cell line) were purchased from the ATCC (Manassas, USA) and grown in Dulbecco’s modified essential medium (Hyclone, Logan, UT, USA) supplemented with 10% fetal bovine serum, 100 units/mL penicillin, and 100 μg/mL streptomycin (Hyclone). Cells were maintained in a humidified 5% CO_2_ atmosphere at 37 °C.

### 2.4. Dual-Luciferase Assay

An antioxidant response element (ARE) promoter fragment (697 bp) of human the NQO1 promoter was extracted from the genomic DNA of HCT116 human colorectal cancer cells and inserted into the pGL3-Basic plasmid (Promega, Madison, WI, USA). HepG2 cells were seeded in a 24-well plate at a density of 6 × 10^4^, incubated for 24 h, and transiently cotransfected with the ARE-Luc reporter plasmid or the control plasmid pRL-SV40 using the TransFast™ reagent (Promega). Transfected cells were treated with 7-ML for 20 h and then with tBHP (*tert*-butyl hydroperoxide; 200 μM) for 2 h. Firefly and Renilla luciferase activities were then determined using a Dual-Luciferase reporter gene assay system (Promega). Nrf2 (nuclear factor-erythroid 2-related factor 2) luminescence signals were normalized versus SV40 (Renilla) luciferase activity. As a positive control for ARE-luciferase activity, sulforaphane was used [33].

### 2.5. Animals

Six-week-old female SKH-1 hairless mice were obtained from the Orientbio (Seongnam, Korea) animal facility and maintained in a controlled environment (25 ± 5 °C, RH 55 ± 5%) under a 12 h light–dark cycle. The animals were provided with sterile food and water ad libitum. All animal experimental procedures were approved beforehand by the KIST Institutional Animal Care and Use Committee (Certification No. KIST-2016-011).

### 2.6. DNCB-Induced Animal Experiments

2,4-Dinitrochlorobenzene (DNCB; Sigma-Aldrich, Seoul, Korea) was used for sensitization and was applied to the dorsal skins of rats at 1% in 200 μL of an acetone/olive oil mix (3:1) once daily for a week. DNCB (0.1% in 200 μL of the acetone/olive oil mixture) was then administered three times weekly for 2 weeks. Alternatively, sensitized mice were treated with 7-ML twice daily for 2 weeks and with DNCB as similarly described above. Positive controls were treated with dexamethasone (0.1% in propylene glycol/EtOH = 7:3) once daily for 2 weeks.

### 2.7. Histological Examination

To assess epidermal thicknesses and inflammatory cell infiltration, dorsal skin lesions were fixed in 10% formalin for 24 h, embedded in paraffin for 24 h, sectioned at 2–3 mm, transferred to slides, and dried overnight. Tissue sections were then stained with hematoxylin and eosin (H&E) or toluidine blue at 37 °C, and epidermal thicknesses and inflammatory infiltration were assessed by observation under an optical microscope (Olympus CX31/BX51, Olympus Optical Co., Tokyo, Japan).

### 2.8. Measurement of Total Serum IgE and IL-4 Levels

Blood samples were collected, centrifuged at 10,000 rpm for 15 min at 4 °C, and stored at −70 °C until required. Serum IgE and IL-4 concentrations were measured using an IgE ELISA kit (eBioscience, San Diego, CA, USA).

### 2.9. Western Blotting

Phosphorylated Nrf2 (pNrf2), Nrf2, and heme oxygenase-1 (HO-1) protein levels were assessed by Western blotting. Briefly, mouse dorsal tissues were lysed in a suitable volume of RIPA buffer (BioPrince, Chuncheon, Korea), protease inhibitor cocktail (Roche, Basel, Switzerland), and phosphatase inhibitor cocktail 2,3 (Sigma-Aldrich, St. Louis, MO, USA). The homogenate obtained was centrifuged at 13,000 rpm for 20 min at 4 °C, and supernatant proteins (20 μg) were separated by SDS-polyacrylamide gel by electrophoresis and transferred to PVDF membranes (Millipore, Billerica, MA, USA). Blots were incubated with primary antibodies against pNrf2 (1/500; Thermo Fisher Scientific, Waltham, MA, USA), Nrf2 (1/500; BioLegend, San Diego, CA, USA), HO-1 (1/500; Abcam, Cambridge, UK), and GAPDH (1/5000; Cell Signaling, Danvers, MA, USA), and then treated with HRP-conjugated secondary antibodies (1/3000; Santa Cruz, CA, USA) and visualized using an ECL kit (Thermo Fisher Scientific). Band densities were determined using the ImageJ (version. 1.5.2, NIH, Bethesda, MD, USA) and normalized versus GAPDH.

### 2.10. Measurement of TEWL and Skin Hydration

The TEWL and skin moisture levels were measured using a Tewameter TM210 (Courage and Khazaka, Cologne, Germany) and a Skin-O-Mat (Cosmomed, Ruhr, Germany), respectively. Measurements were performed weekly under controlled conditions (25 ± 5 °C/RH 55 ± 5%).

### 2.11. Statistical Analysis

The data are expressed as means ± standard deviations (SDs) or as means ± standard errors of means (SEMs). The analysis was conducted by one-way analysis of variance (ANOVA) using the GraphPad Prism 5.0 program (GraphPad Software Inc., San Diego, CA, USA). Statistical significance was accepted for *p*-values < 0.05.

## 3. Results

### 3.1. Effect of 7-ML on tBHP-Induced Nrf2 Transactivation Activity

The effect of 7-ML on Nrf2 transactivation is shown in Figure 2. tBHP reduced Nrf2 expression in HepG2 cells, but this reduction was significantly and dose-dependently suppressed by 7-ML pretreatment (by 12% at 1 μM and 38% at 10 μM) versus tBHP-treated controls.

### 3.2. Histological Evaluation of 7-ML on DNCB-Induced AD-like Skin Lesions

The effects of topical 7-ML treatment on DNCB-induced skin lesions were investigated over the 3-week experimental period. After DNCB sensitization for 1 week (on experimental day 7 (ED7)), severe AD-like symptoms, such as crusts, erythema, hemorrhage, and dryness, were observed on dorsal skin. On ED14 and 22, mice treated with 0.1% or 1% 7-ML showed clinical improvements in AD-like symptoms as compared with DNCB-treated controls (Figure 3). The skin sections collected on ED22 were H&E and toluidine-blue-stained to evaluate epidermal thicknesses and mast cell infiltration, respectively, and the skins of DNCB-treated controls exhibited epidermal hypertrophy and mast cell infiltration. The application of 7-ML at 0.1% or 1% reduced DNCB-induced epidermal thickness to 78 (by 21%) and 62 μm (by 37%), respectively (Figure 4A,C), and topical application of 7-ML at these concentrations reduced mast cell infiltration by 14% and 30%, respectively, versus DNCB-treated controls (Figure 4B,D). Dexamethasone reduced epidermal thickness to 63 μm (by 36%) and reduced mast cell infiltration by 40% as compared with DNCB-treated controls.

### 3.3. Effects of 7-ML on DNCB-Induced Inflammatory Cytokine Expression

The suppressive effects of 7-ML on IL-4 and IgE expressions are shown in Figure 5. Serum IL-4 and IgE levels were higher in DNCB-treated controls than in nontreated controls, but 7-ML treatment at 0.1% and 1% reduced these serum IL-4 levels by 56% and 42%, respectively, as compared with DNCB-treated controls (Figure 5A). In addition, 0.1% and 1% 7-ML significantly reduced serum IgE levels by 26% and 23%, respectively, as compared with DNCB-treated controls (Figure 5B).

### 3.4. Effects of 7-ML on DNCB-Induced Antioxidant Enzyme Expression

The effects of 7-ML on pNrf2, Nrf2, and HO-1 protein levels in mouse skin tissues are shown in Figure 6. DNCB reduced the pNrf2, Nrf2, and HO-1 levels, but cotreatment with DNCB and 7-ML increased pNrf2 levels (by 28%) in the 1% 7-ML group and Nrf2 activity by 40% and 23% in the 0.1% and 1% 7-ML groups, respectively, as compared with DNCB-treated controls (Figure 6B,C). In addition, the expression of the antioxidant gene HO-1 was 57% higher in the 1% 7-ML group than in the DNCB-treated controls (Figure 6D).

### 3.5. Effects of 7-ML on Skin Barrier Function

The effects of 7-ML on TEWL and skin hydration during the 3-week study period are shown in Figure 7. On ED22, the DNCB-treated controls (56.1 g/m^2^/h) showed an increase in TEWL versus nontreated controls (20.7 g/m^2^/h). A reduction in TEWL was observed in the 0.1% 7-ML group (51.9 g/m^2^/h), the 1% 7-ML group (49.5 g/m^2^/h), and the positive (dexamethasone) control group (65.6 g/m^2^/h) (Figure 7A). Skin hydration observed on ED22 was markedly lower in DNCB-treated controls than in nontreated controls. As compared with that of DNCB-treated controls, 1% 7-ML treatment increased skin hydration by 34%. Dexamethasone and 7-ML produced similar results (Figure 7B).

## 4. Discussion

In a previous study, an ethanol extract of *W. ganpi* exhibited antiatopic activity by inhibiting inflammatory factors, such as IL-4, IgE, TNF-α, and IFN-γ, in a mouse model of contact dermatitis [26], and 7-ML (a methylated luteolin isolated from *W. ganpi* extract) inhibited IL-4, IL-6, GM-CSF, and G-SCF in TNF-α-induced HaCat cells [27]. Additionally, it was recently reported that flavonoids exhibit anti-AD effects by inhibiting IL-4 and IgE (known mediators of AD) [16,20], to inhibit oxidative stress, and activate the Nrf2/HO-1 pathway [13,34]. Therefore, our findings that 7-ML acts as a potent antioxidant and anti-inflammatory suggest its potential use as an antiatopic agent.

Nrf2 is a redox-sensitive transcription factor involved in various inflammatory diseases [35]. This transcription factor is activated through the ARE pathway in the presence of oxidative stress and activates the transcriptions of numerous antioxidant genes, including that of HO-1 [36]. The activation of the Nrf2/HO-1 pathway maintains cellular homeostasis in the presence of oxidative stress and inflammation [37,38]. In addition, previous studies on the Nrf2/HO-1 pathway have shown that its activation improves AD-like symptoms by inhibiting the release of inflammatory mediators [39,40]. In the present study, we evaluated the anti-AD effect of 7-ML in DNCB-stimulated SKH-1 mice and its activation of the Nrf2/HO-1 pathway. Our results showed that topic treatment with 7-ML attenuated severe AD-like symptoms, such as erythema, edema, and skin dryness. In addition, 7-ML protected mice from DNCB-induced downregulations of pNrf2, Nrf2, and HO-1. These results demonstrate that 7-ML ameliorates DNCB-induced inflammatory response by activating the Nrf2/HO-1 pathway, thus inhibiting oxidative stress.

Skin inflammatory diseases have common features, such as the overproduction of Th2 cytokines and impaired skin barrier function. IL-4 is a representative Th2 cytokine that induces T cell differentiation and IgE synthesis, activates mast cells, mediates pruritus [41], inhibits keratinocyte differentiation, and thus disrupts the skin barrier [42]. We evaluated the anti-AD effect of 7-ML on IL-4, IgE, and skin barrier function in DNCB-induced SKH-1 mice, which is an accepted study model of AD [43]. 7-ML treatment reduced epidermal thickness and mast cell infiltration and improved AD-like skin lesions in the dorsal skin of hairless mice. In addition, DNCB-induced increases in serum IL-4 and IgE levels were suppressed by 7-ML at concentrations of 0.1% and 1%. Furthermore, 7-ML reduced DNCB-induced TEWL increases and skin hydration decreases. These observations show that by inhibiting IL-4 and IgE synthesis, 7-ML ameliorated DNCB-induced skin barrier dysfunction and atopic itching in our mouse model of AD.

Collectively, 7-ML was found to inhibit the activations of IL-4 and IgE and mast cell activation, which are major mediators of the pathogenesis and development of AD, and suppressed DNCB-induced skin barrier dysfunction in our mouse model. In addition, 7-ML suppressed atopic inflammatory response by activating the Nrf2/HO-1 pathway and, thus, reduced oxidative stress in DNCB-treated mouse skin tissues. Our findings show that 7-ML should be considered a potential antiatopic agent with potent inflammatory and antioxidant effects. 

## 5. Conclusions

We evaluated the in vitro and in vivo antioxidative and anti-AD effects of 7-O-methylluteolin isolated from *W. ganpi*. 7-ML was found to activate the Nrf2/HO-1 signaling pathway in tBHP-stimulated HepG2 cells and DNCB-stimulated SKH-1 hairless mice. Furthermore, topical application of 7-ML to mice suppressed the DNCB-induced upregulations of IL-4 and IgE and improved skin barrier functions. We suggest that 7-ML offers an attractive starting point for the development of AD preventative and therapeutic agents.

## Figures and Tables

**Figure 1 antioxidants-11-01344-f001:**
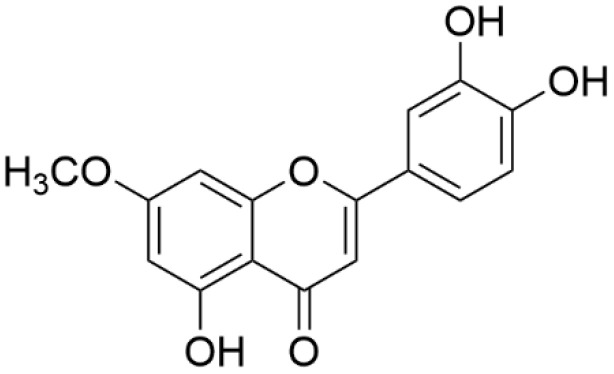
Chemical structure of 7-*O*-methylluteolin.

**Figure 2 antioxidants-11-01344-f002:**
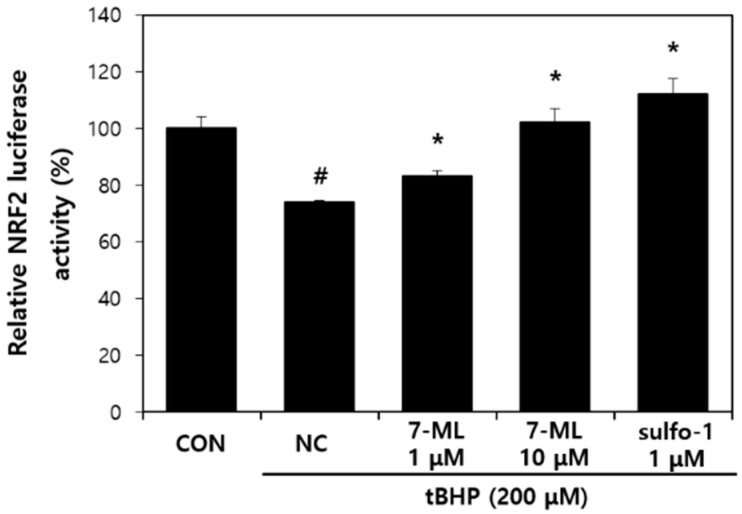
Effect of 7-ML on Nrf2 transactivation. The pGL3-Basic-ARE plasmid was transiently transfected into HepG2 cells and treated with 7-ML for 20 h after transfected, and tBHP (200 μM) with 7-ML was further treated for 2 h, after which luciferase activity was measured. Luciferase activities were normalized versus nontreated controls. Results are expressed as means ± SDs (*n* = 3). ^#^ *p* < 0.05 vs. nontreated cells; * *p* < 0.05 vs. the tBHP-treated controls. CON, nontreated control; NC, tBHP-treated negative control; 1 or 10 μM 7-ML, tBHP + 1 or 10 μM 7-ML-treated groups; 1 μM sulfo-1, tBHP + sulforaphane-treated group.

**Figure 3 antioxidants-11-01344-f003:**
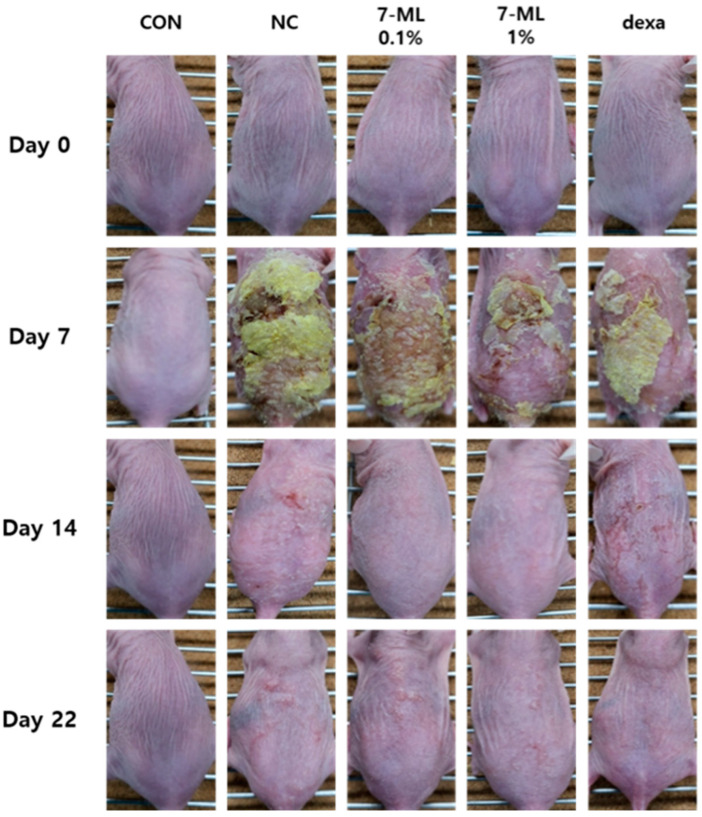
Effects of 7-ML on atopic dermatitis-like symptoms in the DNCB-induced mouse model. Photographs of the AD-like skin symptoms exhibited by SKH-1 hairless mice were evaluated during the 3-week experimental period. CON, nontreated control; NC, DNCB-treated negative control; 0.1% or 1% 7-ML, DNCB + 0.1% or 1% 7-ML-treated groups; dexa, DNCB + dexamethasone-treated group.

**Figure 4 antioxidants-11-01344-f004:**
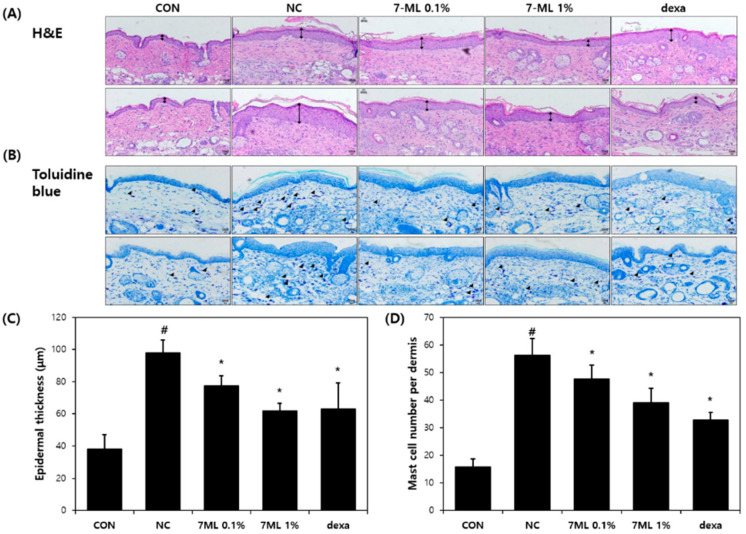
Effects of 7-ML on epidermal thickness and mast cell numbers in the DNCB-induced mouse model. Dorsal skin sections were harvested and stained with H&E (**A**) or toluidine blue (**B**) to evaluate epidermal thicknesses (**C**) and mast cell numbers (**D**). Black up–down arrow indicates epidermis. Black arrow head indicates mast cell. The results are expressed as means ± SDs (*n* = 3). ^#^ *p* < 0.05 vs. nontreated controls; * *p* < 0.05 vs. DNCB negative controls. CON, nontreated control; NC, DNCB-treated negative control; 0.1% or 1% 7-ML, DNCB + 0.1% or 1% 7-ML-treated groups; dexa, DNCB + dexamethasone-treated group.

**Figure 5 antioxidants-11-01344-f005:**
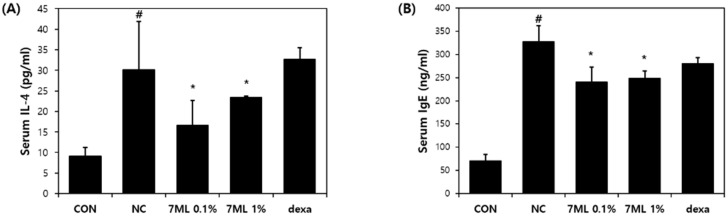
Effects of 7-ML on serum IL-4 and IgE levels in DNCB-induced mice. Serum IL-4 (**A**) and IgE (**B**) levels were measured using an ELISA kit. The results are expressed as means ± SDs (*n* = 3). ^#^ *p* < 0.05 vs. nontreated controls; * *p* < 0.05 vs. DNCB negative controls. CON, nontreated control; NC, DNCB-treated negative control; 0.1% or 1% 7-ML, DNCB + 0.1% or 1% 7-ML-treated groups; dexa, DNCB + dexamethasone-treated group.

**Figure 6 antioxidants-11-01344-f006:**
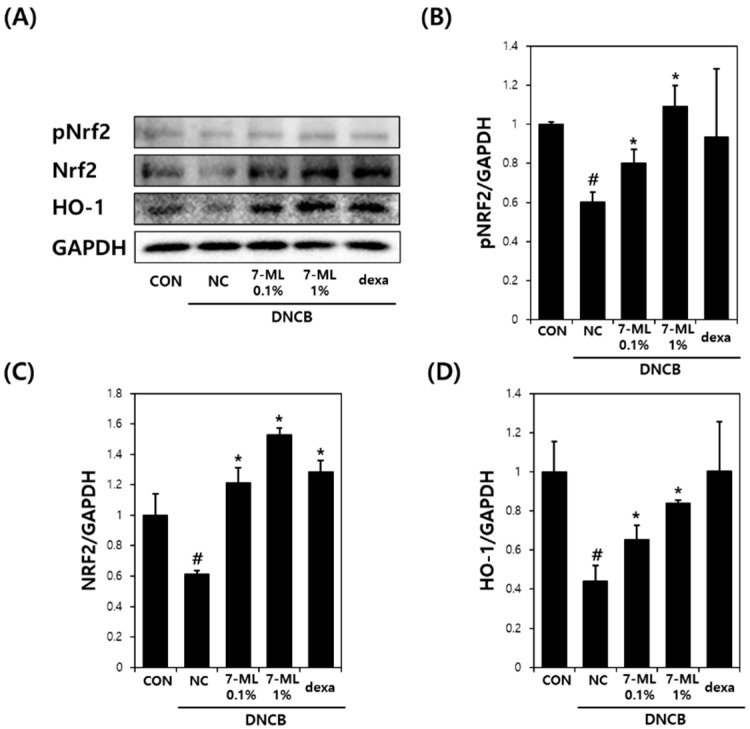
Effects of 7-ML on Nrf2/HO-1 pathway activation in a DNCB-induced mouse model. pNrf2, Nrf2, and HO-1 protein levels in mouse skin tissues were measured by Western blotting (**A**). pNrf2, Nrf2, and HO-1 levels were normalized versus GAPDH (**B**–**D**). Results are expressed as means ± SEMs (*n* = 3). ^#^ *p* < 0.05 vs. nontreated controls; * *p* < 0.05 vs. DNCB controls. CON, nontreated control; NC, DNCB-treated negative control; 0.1% or 1% 7-ML, DNCB + 0.1% or 1% 7-ML-treated groups; dexa, DNCB + dexamethasone-treated group.

**Figure 7 antioxidants-11-01344-f007:**
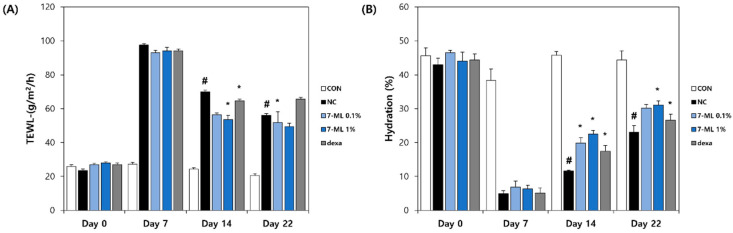
Effects of 7-ML on skin barrier function in a DNCB-induced mouse model. The level of TEWL (**A**) and the level of skin hydration (**B**) were measured using a Tewameter TM210 and Skin-O-Mat for 3 weeks. The results are expressed as a mean ± SD (*n* = 3). ^#^ *p* < 0.05 vs. nontreated control; * *p* < 0.05 vs. the DNCB-treated negative control. CON, nontreated control; NC, DNCB-treated negative control; 0.1% or 1% 7-ML, DNCB + 0.1% or 1% 7-ML-treated groups; dexa, DNCB + dexamethasone-treated group.

## Data Availability

The data are contained within the article.

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
