# Peer review of "7-O-Methylluteolin Suppresses the 2,4-Dinitrochlorobenzene-Induced Nrf2/HO-1 Pathway and Atopic Dermatitis-like Lesions"

_antioxidants, 2022, doi:10.3390/antiox11071344_

Round 1

Reviewer 1 Report

The authors evaluated the in vitro and in vivo antioxidative and anti-AD effects of 7-O-methylluteolin isolated from Wikstroemia ganpi.

It is an interesting topic. But there are some things that should be clarified or modified in the manuscript.

1.      Supply the related research status about 7-O-methylluteolin, including which plants have been isolated from and its activities in the Introduction part.

2.      L81 Add the extraction temperature.

3.      Please check the position of C-3′ hydroxy in the chemical structure in Figure 1.

4.      What is the statistical software mentioned in 2.11 on L166 ?

5.      The figure notes in Figure 2, 0.1% or 1% 7-ML; tBHP + 0.1% or 1% 7-ML-treated groups, do not match the expression figures for 7-ML 1 μM, 7-ML 10 μM. Please modify or specify.

6.      Figure notes annotation are unclear in Figure 3-7. For example 0.1% or 1% 7-ML, DNCB + 0.1% or 1% 7-ML- treated groups.

7.      The area of change should be marked in Figure 4 of Fig. A and Fig. B.

Author Response

1. Supply the related research status about 7-O-methylluteolin, including which plants have been isolated from and its activities in the Introduction part.

-  I appreciate reviewer’s valuable comment. We newly added the part such as’ The 7-ML has been isolated from diverse plants such as W. ganpi, Daphne oleoides, Avicennia marina, Coleus parvifolius, and Thymus vulgaris [27-31] and has been reported to have anti-cancer [29], anti-HIV [30], and antioxidant [31] effects.’ to the Introduction in revised version of manuscript and please see yellow-highlighted part (line 69-72).

2. L81 Add the extraction temperature.

-  We now added the extraction temperature (line 83-84).

3. Please check the position of C-3′ hydroxy in the chemical structure in Figure 1.

-  The C-3’ hydroxy position is now corrected and please see the chemical structure in Figure 1.

4. What is the statistical software mentioned in 2.11 on L166 ?

-  Statistical analysis was performed using the GraphPad Prism 5.0 program and software name is newly inserted to the text (line 170).

5. The figure notes in Figure 2, 0.1% or 1% 7-ML; tBHP + 0.1% or 1% 7-ML-treated groups, do not match the expression figures for 7-ML 1 μM, 7-ML 10 μM. Please modify or specify.

-  It is now changed to '1 μM or 10 μM 7-ML, tBHP + 1 μM or 10 μM 7-ML-treated groups' (Figure 2) and the order of the experimental group is changed to CON > NC > 1 μM 7-ML > 1 μM 7-ML > dexa.

6. Figure notes annotation are unclear in Figure 3-7. For example 0.1% or 1% 7-ML, DNCB + 0.1% or 1% 7-ML- treated groups.

-  To clarify, the Figure notes annotation is now changed to '0.1% or 1% 7-ML, DNCB + 0.1% or 1% 7-ML- treated groups' (Figure 3-7) and the order of the experimental group is changed to CON > NC > 0.1% 7-ML > 1% 7-ML > dexa.

7. The area of change should be marked in Figure 4 of Fig. A and Fig. B.

- It is now inserted as ‘Black up-down arrow indicate epidermis. Black arrow head indicate mast cell.’ (line 210-211).

Reviewer 2 Report

The study titled “7-O-Methylluteolin Suppresses the 2,4-Dinitrochlorobenzene-Induced Nrf2/HO-1 Pathway and Atopic Dermatitis-Like Lesions” aimed to investigate the potential of 7-O-methyl luteolin as a bioactive compound against atopic dermatitis, based on previous results with plant extracts in vivo and with the pure compound in vitro, in a keratinocyte model. Overall, the study the is well designed and the data clearly support the hypothesis. The results are suitable for publication, however, some suggestions are listed below for consideration:

1)     Dual Luciferare assay (lines 114, 115 versus lines 176, 177) – clearly state how cells were exposed, the two segments of text seem contradictory. It is not straightforward, so 20h after transfection, cells were incubated with 7-ML for 2h, then the medium was replaced with medium containing tBHP and cells were incubated for an additional 2h period? Or were the cells incubated 20h with 7-ML, the medium and removed and cells were incubated 22h with tBHP? The rationale for using 1 and 10 micromolar 7-ML in these experiments is missing.

2)     Sulforaphane control (line 180) – clearly state where the conditions used (SFN concentration and exposure time) come from, for it to be used as positive control

3)     pNrf2 levels (lines 226-228) – Nrf2 activation is a multistep process. Nrf2 phosphorylation and Nrf2 activation, although showing some overlap, are distinct concepts. The sentence would be better rephrased as “increased pNrf2 levels”.

4)     Western blot image versus normalized levels – The quantification relative levels of pNfr2, total Nfr2 does not seem to be reflected by the blot provided. GAPDH bands are not very different between samples. Bands of pNfr2 appear most intense in the order: CON > 7ML 1% > 7 ML 0.1% > DEXA > NC, but this is not the result presented in Fig. 6B. Also, bands of total Nrf2 follow the trend: DEXA > 7-ML 1% > 7-ML 0.1% > CON > NC.
